# The Emotional Response to Pandemic of Middle- and High-School Students of an Italian Northern Province: The ERP Study

**DOI:** 10.3390/children9010059

**Published:** 2022-01-04

**Authors:** Massimiliano Povero, Paola Turco, Roberto Walter Dal Negro

**Affiliations:** 1AdRes Health Economics and Outcome Research, 10121 Torino, Italy; 2Research & Clinical Governance, 37100 Verona, Italy; turcop@libero.it (P.T.); robertodalnegro@gmail.com (R.W.D.N.)

**Keywords:** COVID-19 pandemic, emotional reactions, teenagers, students

## Abstract

The COVID-19 outbreak variably affected people’s mental reactions worldwide but was only episodically investigated in healthy Italian teenagers. Our aim was to investigate the emotional responses of Italian middle and high school students to the pandemic. An anonymous 10-item questionnaire was distributed in pre-selected school samples. Responders had to score their perceived extent for each reaction from 0 (lowest perception) to 10 (highest perception). A group of adults was selected as control. Generalized linear models were used to estimate differences among adults and students, high school (HS) and middle school (MS) students, and urban (U) and rural (R) MS students. Comparisons were presented as mean difference (Δ) with a 95% confidence interval (CI). A total of 1512 questionnaires (635 adults, 744 HS, 67 UMS, and 66 RMS) were analyzed. Students appeared more indifferent (Δ = 1.97, 1.52–2.41), anxious (Δ = 0.56, 0.07–1.04), aggressive (Δ = 2.21, 1.72–2.70), and depressed (Δ = 1.87, 1.40–2.34) than adults did, and claimed a higher loss of interest in their activities (Δ = 1.21, 0.72–1.70). Students were less disbelieving (Δ = −0.93, −1.50–0.35) and feared for their loved ones (Δ = −0.89, −1.40–0.39). MS students were less affected by the outbreak than HS students were. Furthermore, R-MS students were significantly less aggressive and depressed, but more indifferent and disbelieving than U-MS. Female sex was an independent factor associated to almost all the questionnaire domains. The pattern of the psychological responses to the pandemic in Italian students proved multifaceted. In addition to anxiety, loss of interest in activities, and depression, aggressiveness emerged as the most characterizing mental attitude in response to the pandemic.

## 1. Introduction

The long-lasting derangement of usual activities and social relationships, together with the loneliness induced by the COVID-19 pandemic, variably impacted people’s mental reactions worldwide [1].

The mental impact of the COVID-19 outbreak was widely investigated in general populations; however the majority of studies available in the literature mainly focused on specific clusters of adult subjects identified at higher risk: the elderly [2,3,4], pre-existing psychiatric patients [5,6,7,8,9,10], health care workers [11,12,13,14,15], persons in detention [16,17], migrants [18], and university students [19,20,21]. Moreover, the few Italian studies presently available concentrated their attention on the emotional impact of the COVID-19 pandemic in adults, mostly [22,23,24,25].

After the first Chinese reports [26], limited attention was paid to the different psychological effects of the pandemic in adolescents and middle school students [27,28,29,30,31,32,33,34], particularly in western countries [34,35]. In general, the international literature reported a higher prevalence of anxiety and depressive symptoms in adolescents [34,35,36], particularly in females [27,28,29,30,31,32,33,34,35,36,37,38], together with a high risk of their negative coping [26].

In Italy, the psychological distress due to the COVID-19 pandemic was only episodically investigated in healthy teenagers. A significant emotional dysregulation was generically described during the early Italian phase of the 2019 pandemic, and it was associated with an increased vulnerability to the stressful effect of the coronavirus outbreak [37]. A descriptive Italian cross-sectional report, though carried out without any support of psychometric questionnaires or scales, tended to minimize this aspect, and rather emphasized the excellent ability of adolescents to manage the adverse emotional conditions and situations of insecurity caused by the pandemic [38]. Recently, a cross-sectional study enrolled two cohorts of Italian subjects (for a total of more than of 1100 records), investigated the relationship between post-traumatic stress disorder symptoms (i.e., intrusion, hyperarousal, and avoidance) and the fear of COVID-19 and mental health [39]. The results showed that within the post-traumatic stress disorder framework, hyperarousal compared with avoidance mediated the relationship between intrusion and the analyzed outcomes.

Stemming from the evidence provided by the few Italian surveys currently available, the present study was planned with the aim to investigate the emotional response of Italian middle and high school students to the COVID-19 pandemic (the ERP study) and to highlight possible differences in terms of emotional response by age and sex.

## 2. Materials and Methods

The investigational tool adopted in the study was an anonymous form reporting ten different domains of possible emotional reactions induced by the pandemic. Responders were merely required to score their perceived extent for each reaction in a Likert scale from 0 (lowest perception) to 10 (highest perception). The list of possible emotional reactions was based on a previous systematic literature review [31] and was purposely reported in a scattered order. As aggressiveness was never investigated in all other studies but it could be relevant when dealing with adolescents, it was also included in the questionnaire. Further variables requested from responders were their age, sex, and the location of their school institute (city or rural location) (Figure 1).

The questionnaire had been previously validated in a different sample of adolescents during the pre-study phase, and its comprehension resulted in 100%. Once the study was accepted by the Heads of the Institutes selected and by the teachers voluntarily involved, the questionnaire was distributed to the students who were invited to fill the form completely. Data collection was carried out online via Google forms in 96.4% of cases and the remaining 3.6% by paper questionnaires.

The study was conducted in the Province of Verona (a Province highly affected by the coronavirus outbreak over the last eighteen months) during May-June 2021, before the summer closure of schools. A sample of high school (HS) and middle school (MS) institutes was randomly selected. As high school institutes are concentrated in the cities, while middle schools are much more widespread within the Province, the students’ sample was proportionally calculated for urban (U) and rural (R) schools. Assuming that adolescents’ emotional reactions would likely be different from those of adult individuals, a pre-calculated sample of adults was also included in order to compare their pattern of response to that of adolescents. Questionnaires were administered, over the same period, to people working in or linked to the schools involved in the study, and to parents and relatives of students who accepted to participate in the study.

Based on the preliminary analysis on the first 100 questionnaires collected, a 30% difference between adults’ and adolescents’ responses was considered clinically significant. The formula for two independent (unmatched) samples was used (59): with 5% type I error and 90% statistical power, and assuming a 20% dropout rate, at least 756 patients (378 adults and 378 students) should be enrolled in the study [40].

Continuous data were presented as means and standard deviation (SD), while categorical data as absolute and relative frequencies. Median and interquartile range were also reported for the questionnaire scores. Differences in baseline characteristics among the groups recruited were tested by non-parametric Wilcoxon test for continuous data and by Fisher exact test for categorical variables.

Each domain of the questionnaire was analyzed by a generalized linear regression model (family Gamma). The difference between adults and students (in the preliminary analysis), between HS and MS students, and between U-MS and R-MS students (in the secondary analysis) were adjusted by age and sex. Moreover, a global analysis was performed considering a 4-level categorical variable (0: adults; 1: HS students; 2: U-MS students, and 3: R-MS students) in order to evaluate all the differences among the four subgroups. Comparisons were presented as mean difference (Δ) together with the corresponding 95% confidence interval (95% CI). To correct for multiple comparison, in the last analysis, 99.15% CI were calculated according to Šidák correction [40].

## 3. Results

A total of 1530 questionnaires were collected, by a redemption of 98.2%: 887 among students and 643 among adults. Eighteen questionnaires were excluded due to missing or wrong information reported (ten among students and eight among adults). Finally, 1512 questionnaires (877 students and 635 adults) were included in the study, and then analyzed. The internal reliability of the variables considered in the questionnaire was adequate and consistency was high (Cronbach’s alpha = 0.7995).

Baseline characteristics were reported in Table 1. Sex was quite balanced among all subgroups, with the exception of that of HS students among whom the percentage of females was higher (70.43%, Fisher exact test, *p* < 0.001). Moreover, no difference was detected in the age distribution of students from U-MS and R-MS institutes (12.7 vs. 12.3 years of age, *p* = 0.1704).

According to the primary analysis (Table 2 and Appendix A), when compared to adults, students appeared more indifferent to the pandemic (Δ = 1.97, 95% CI 1.52 to 2.41), though more anxious (Δ = 0.56, 95% CI 0.07 to 1.04), more aggressive (Δ = 2.21, 95% CI 1.72 to 2.70), more depressed (Δ = 1.87, 95% CI 1.40 to 2.34), and also claiming a much higher loss of interest in their activities (Δ = 1.21, 95% CI 0.72 to 1.70). On the contrary, students were less disbelieving (Δ = −0.93, 95% CI −1.50 to −0.35) and claimed less fear for their loved ones (Δ = −0.89, 95% CI −1.40 to −0.39). Irritation, concern, and fear for themselves were similar among students and adults.

In general, MS students appeared less affected by the COVID-19 outbreak than HS students (Table 2 and Appendix A). In particular, they experienced significantly less irritation (Δ = −1.28, 95% CI −1.94 to −0.63), anxiety (Δ = −1.22, 95% CI −1.91 to −0.53), aggressiveness (Δ = −2.76, 95% CI −3.39 to −2.12), concern (Δ = −0.88, 95% CI −1.52 to −0.24), and depression (Δ = −1.79, 95% CI −2.53 to −1.06). In particular, students of rural areas were significantly less aggressive (Δ = −1.65, 95% CI −2.81 to −0.48), less depressed (Δ = −1.64, 95% CI −2.87 to −0.40), and more indifferent (Δ = 138, 95% CI 0.20 to 2.56) and disbelieving (Δ = 1.51, 95% CI 0.35 to 2.67) than those of urban schools (Table 2 and Appendix A).

Female sex proved to be an independent, critical factor associated with almost all domains investigated by the questionnaire (Figure 2).

Specifically, females were more irritable (Δ = 0.35, 95% CI 0.02 to 0.68), anxious (Δ = 1.73, 95% CI 1.41 to 2.05), concerned (Δ = 1.29, 95% CI 1.01 to 1.57), afraid for themselves (Δ = 1.69, 95% CI 1.38 to 2.00) and for their loved ones (Δ = 1.27, 95% CI 0.98 to 1.55), and depressed (Δ = 0.74, 95% CI 0.41 to 1.06). Equivalent trends were observed for females in all the four subgroups (see Appendix A).

Multiple comparison analysis involving the 4-level categorical variable (0: adults; 1: HS students; 2: U-MS students, and 3: R-MS students) confirmed previous results (Table 3). In particular, students were more disbelieving and less fearful for their loved ones than adults, regardless of their school level (high or middle) and geographic area (urban or rural). When compared to adults, the highest levels of aggressiveness and depression were mostly recorded in HS students. In general, MS students were significantly less irritated, anxious, aggressive, indifferent to their activities, and depressed than high school students, regardless of their school location (urban or rural). U-MS students and adults were actually comparable, while R-MS students were significantly less aggressive and depressed.

Results of the correlation analysis carried out within the 10 items of the questionnaire was reported in Table 4. Irritation, anxiety, aggressiveness, concern, and fear were strongly correlated with each other (*p* < 0.001), regardless of the subgroup analyzed (see Appendix A). Moreover, a strong linear association was observed between irritation vs. anxiety/aggressiveness, anxiety vs. concern/fear for themselves, and concern vs. fear for themselves and their loved ones (see Appendix A).

## 4. Discussion

The COVID-19 outbreak disrupted the behaviors and attitudes of many communities worldwide [1]. In general, the pandemic has been described as capable of causing higher levels of psychopathological symptoms in adults regardless of the study design adopted, though reactions appeared variably characterized in different sets of subjects [2,3,4,5,6,7,8,9,10,11,12,13,14,15,16,17,18,19,20,21,22,23,24,25].

The psychological reactions of children and adolescents to the pandemic were investigated by traditional methods and social media platforms in different studies [25,26,27,28,29,30,31,32,33,34,35,36,37,38]. The majority of studies agreed in affirming the pandemic as capable of significantly affecting the psychological profile of teenagers, even in the absence of prior mental disorders. Anxiety, panic attacks, irritability, and depression were the most frequent mood disorders reported in these cases [31,33,41], with a prevalence of approximately 20% in adolescents [26].

The majority of the subjects investigated in a study conducted on 1067 students aged 6–18 years, from 56 schools located, proved to be psychologically affected by the pandemic, females in particular. Their psychological impact, mostly characterized by panic reactions, was much more pronounced than the physical impact (78.1% vs 1.3%, respectively; *p* < 0.001) [27]. A further survey reported the occurrence of behavioral disorders and emotional (such as anxious or anxious-depressive) symptoms in children and adolescents [34].

A large online survey confirmed these results and added evidence that ethnicity might variably affect the occurrence of outbreak-induced mental disorders in young people in terms of anxiety, lower vitality, increased depression, and lower self-control [33]. As expected, subjects with prior mental health disorders and/or belonging to disadvantaged families showed a clear worsening in their mental health status with higher probability (OR = 1.68 and 5.83, respectively) [33]. A recent review on near 23,000 children and adolescents from 15 studies further confirmed that a significant proportion of them was suffering from anxiety (34.5%), depression (41.7%), irritability (42.3%), and inattention (30.8%) [31].

In Italy, the stressful impact of the pandemic was investigated for the first time in 500 healthy children during the early Italian phase of the COVID-19 outbreak [37]. Concerns and fears induced by the pandemic were further investigated in an unspecified number of secondary school-first degree students [38]. Even if the study was merely descriptive and biased by the non-use of any psychometric scale or questionnaire, the data suggested the tendency of these young students (in particular males) to underestimate the danger. They seemed less anxious than their parents did; their apprehension was age-dependent, and higher in students from central-southern Italian regions.

The present epidemiological survey, aimed to investigate adolescents’ emotional reactions to the pandemic and the possible occurrence of a response able to discriminate their reactions from those of adults, allowed for some novel and peculiar results. In concordance with a couple of previous studies [38,41], the general psychological pattern of the responses of students to the pandemic confirmed different characterization when compared to that of adults. Furthermore, a multifaceted response was for the first time described within the students’ population, and its age-dependency resized.

High school students were dramatically much more irritated, anxious, concerned, fearful, depressed, and aggressive than adults, peculiarly in the female sex. Moreover, middle school students showed a dichotomous pattern of response: those of the city schools, even if younger, showed a response very similar to that of the high school students, while the response of those of the rural schools was very similar to that of adults, particularly in terms of anxiety, aggressiveness, depression, and loss of interest in their activities.

Different from all previous studies, the present survey also focused for the first time the peculiar degree of aggressiveness induced by the pandemic in middle and high school students, and calculated the linear relationships existing among the different emotional symptoms, variably associated. Actually, aggressiveness mostly discriminated the students’ from the adults’ response, and differently characterized the emotional response within the students’ cohort.

If age can be hypothesized as the critical factor from this point of view, this hypothesis is very difficult to sustain for explaining the difference in the psychological response recorded within rural and city middle school students that are equally aged. Moreover, the results of the present study confirmed that the female sex is characterized by dramatically higher degrees of mental stress, regardless of the age (and the kind of school).

Other factors, still scarcely investigated [29], could likely have to be invoked in order to explain the much smoother response of the former cohort, such as: the hormonal imprinting, a less stressful lifestyle that is much more prone to personal relationships in the smaller communities where they usually live, a less compulsive addiction to social media applications, the lower impact of social media in the spreading of panic reactions. However, it should also be considered the variable abilities of teenagers to manage situations of insecurity and to cope with unfavorable adverse conditions, reported as excellent in some studies [38,40], but insufficient in others [26,27,28,29,30,31,32,33,34,35,36,37]. Obviously, students with higher levels of resilience are more likely to be characterized by positive mental reactions [37].

All the emotional reactions described above, included sleep disorders [31,32,33,34,35,36,37,38,41], can obviously substantially impact young students’ lifestyles and social relationships. As human nature requires social connections [42], the impact of homestay, loneliness, and long-term social isolation can frequently lead to frustration and anger [43], particularly in teenagers as shown in the present study. This aspect represents a peculiar issue that should be regarded as a transversal medical problem of crucial clinical importance. In effect, even though in the absence of prior psychological disorders, young subjects who are suffering from any chronic disease (such as, bronchial asthma, diabetes, cystic fibrosis, epilepsy, hematological disorders, etc.) and need the daily consumption of specific drugs at regular time intervals have a much higher probability to skip doses, to drop significantly the adherence to their treatments, and to reduce dramatically the effectiveness of their therapeutic strategy. Actually, the exalted boredom, depression, and aggressivity induced by the pandemic in adolescents can magnify their negative approach to all procedures requiring regular actions, including the compliance to pharmacological treatments. In this context, a psychological profile mainly characterized by depression, loss of interest, and aggressiveness as the major emotional response, particularly in the female sex, represents a clear risk factor [44].

The present paper has some limitations. First, the study only attains to the secondary school population of a northern Italian province, one heavily affected by the outbreak. Second, prior psychological disorders of students and of the social status of their families were impossible to investigate due to strict privacy limitations. Lastly, the cross-sectional design used in this analysis does not allow for causal inference about the relationship among the study variables. However, the empirical evidence and the differences observed among the enrolled cohorts are plausible and realistic.

Points of strength include: (1) the use of a validated mental scale, (2) the strict epidemiological model adopted that allowed to obtain a representative sample to investigate, (3) the adoption of strict and proper statistical models, (4) the comparison between teenagers and adults, (5) the analytical comparison between high school and middle school students, and (6) the comparison within the middle school students according to the rural and city communities of students.

## 5. Conclusions

The results of the present study highlighted the significant rates of psychological consequences of the pandemic in adolescent and teenager students, and revealed for the first time the differences in their multifaceted pattern of response. In addition to anxiety, a loss of interest in activities, depression, and aggressiveness emerged as the most characterizing mental attitudes in response to the pandemic. The students’ age, but mostly female sex, were the drivers.

As it is not known how the students may react to a persisting pandemic, their psychological profile should be periodically investigated by means of the simple questionnaire used in the present study, and subjects at risk be monitored. There is urgent need to plan (or improve) psychological interventions and mental health support in schools and social institutions in order to reduce and minimize the impact of the outbreak on adolescent and teenage students, and to facilitate their coping with difficulties.

## Figures and Tables

**Figure 1 children-09-00059-f001:**
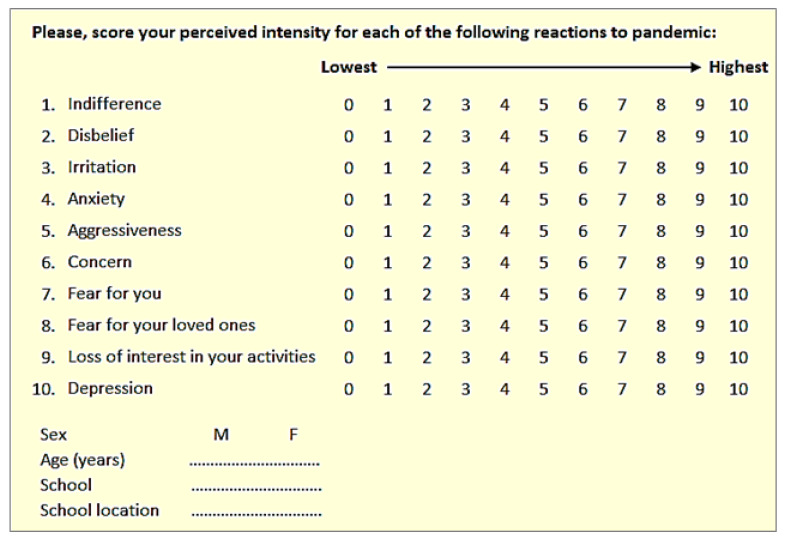
The questionnaire with all variables requested.

**Figure 2 children-09-00059-f002:**
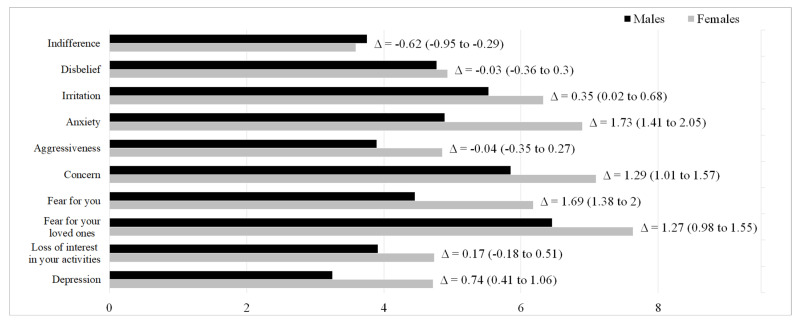
Difference among males and females regarding the 10 questions of the questionnaire in the whole sample (comparison was adjusted by age).

**Table 1 children-09-00059-t001:** Baseline characteristics of the whole sample by subgroups.

Variables	Adults	HS	U-MS	R-MS
N	635	744	67	66
Sex (% males)	303 (47.72%)	220 (29.57%)	32 (47.76%)	34 (51.52%)
Age (mean ± SD)	46.84 ± 17.16	16.71 ± 1.53	12.70 ± 1.85	12.33 ± 1.14

HS: High School; MS: Middle School; U-MS: Urban Middle School; R-MS: Rural Middle School; SD: standard deviation.

**Table 2 children-09-00059-t002:** Baseline characteristics of the whole sample by subgroups: data for each group are reported as mean ± SD, median and range (in brackets); data in bold indicate the most significant difference between adults and students in terms of their emotional response.

Reaction	Adults (n = 635)	Students (n = 877)	Δ (95% CI)	HS (n = 744)	MS (n = 133)	Δ (95% CI)	U-MS (n = 67)	R-MS (n = 66)	Δ (95% CI)
Indifference	2.72 ± 3.11 1 (0–5)	4.33 ± 3.10 4 (1–7)	**1.97 (1.52 to 2.41)**	4.39 ± 3.09 5 (2–7)	3.97 ± 3.18 4 (1–6)	−0.57 (−1.31 to 0.18)	3.33 ± 3.34 3 (0–6)	4.62 ± 2.89 4 (1–6)	**1.38 (0.20 to 2.56)**
Disbelief	5.04 ± 3.20 5 (2–8)	4.75 ± 3.04 5 (2–7)	**−0.93 (−1.50 to −0.35)**	4.94 ± 2.99 5 (2–7)	3.65 ± 3.12 3 (1–6)	−0.50 (−1.25 to 0.24)	2.97 ± 3.11 2 (0–5)	4.35 ± 2.99 4.5 (2–7)	**1.51 (0.35 to 2.67)**
Irritation	5.38 ± 3.17 6 (3–8)	6.47 ± 3.11 7 (4–9)	−0.12 (−0.65 to 0.40)	6.73 ± 3.02 8 (5–9)	5.01 ± 3.17 5 (2–8)	**−1.28 (−1.94 to −0.63)**	5.21 ± 3.55 6 (2–8)	4.80 ± 2.74 5 (3–7)	−0.17 (−1.25 to 0.91)
Anxiety	5.44 ± 3.09 6 (3–8)	6.60 ± 3.18 7 (5–10)	**0.56 (0.07 to 1.04)**	6.89 ± 3.04 8 (5–10)	4.99 ± 3.48 6 (2–8)	**−1.22 (−1.91 to −0.53)**	5.10 ± 3.71 6 (2–9)	4.88 ± 3.27 6 (2–7)	0.04 (−1.21 to 1.29)
Aggressiveness	2.74 ± 2.89 2 (0–5)	5.74 ± 3.42 6 (3–9)	**2.21 (1.72 to 2.70)**	6.26 ± 3.22 7 (3–9)	2.87 ± 3.07 2 (0–5)	**−2.76 (−3.39 to −2.12)**	3.74 ± 3.37 4 (0–6)	1.98 ± 2.46 1 (0–3)	**−1.65 (−2.81 to −0.48)**
Concern	6.81 ± 2.71 8 (5–9)	6.46 ± 2.75 7 (5–9)	−0.31 (−0.79 to 0.17)	6.63 ± 2.60 7 (5–9)	5.53 ± 3.33 6 (3–9)	**−0.88 (−1.52 to −0.24)**	5.39 ± 3.61 6 (1–9)	5.68 ± 3.04 6 (4–8)	0.00 (−1.20 to 1.20)
Fear for you	5.39 ± 3.11 6 (3–8)	5.60 ± 3.09 6 (3–8)	0.29 (−0.23 to 0.81)	5.75 ± 3.00 6 (3–8)	4.72 ± 3.43 5 (1–8)	−0.29 (−0.98 to 0.40)	4.30 ± 3.21 4 (1–7)	5.15 ± 3.62 6 (1–8)	0.74 (−0.45 to 1.94)
Fear for YLO	7.49 ± 2.65 8 (6–10)	6.95 ± 2.91 8 (5–9)	**−0.89 (−1.40 to −0.39)**	7.04 ± 2.77 8 (5–9)	6.46 ± 3.54 8 (3–10)	−0.55 (−1.24 to 0.14)	6.19 ± 3.81 8 (2–10)	6.73 ± 3.25 8 (5–10)	0.61 (−0.64 to 1.87)
LOI in your activities	3.38 ± 3.12 3 (0–6)	5.17 ± 3.45 5 (2–8)	**1.21 (0.72 to 1.70)**	5.42 ± 3.42 6 (2–8)	3.77 ± 3.25 3 (0–6)	−0.40 (−1.20 to 0.40)	3.88 ± 3.53 3 (0–6)	3.67 ± 2.97 3 (1–6)	0.40 (−0.84 to 1.63)
Depression	2.88 ± 2.97 2 (0–5)	5.06 ± 3.56 5 (1–8)	**1.87 (1.40 to 2.34)**	5.44 ± 3.44 6 (2.5–8)	2.93 ± 3.48 1 (0–5)	**−1.79 (−2.53 to −1.06)**	3.74 ± 4.02 2 (0–8)	2.11 ± 2.60 1 (0–4)	**−1.64 (−2.87 to −0.40)**

HS: High School; LOI: loss of interest; MS: Middle School; U-MS: Urban Middle School; R-MS: Rural Middle School; SD: standard deviation; YLO: your loved ones.

**Table 3 children-09-00059-t003:** Multiple comparisons for the 10 items investigated by the questionnaire (data in bold indicate the most significant difference between adults and students in terms of their emotional response).

Reaction	Δ (99.15% CI) Row vs. Column	Adults	HS	U-MS
Indifference	HS	**2.04 (1.41 to 2.66)**		
	U-MS	0.89 (−0.22 to 2.01)	**−1.14 (−2.21 to −0.08)**	
	R-MS	**2.13 (0.66 to 3.61)**	0.10 (−1.34 to 1.54)	1.24 (−0.46 to 2.94)
Disbelief	HS	**−0.82 (−1.61 to −0.02)**		
	U-MS	**−2.88 (−3.90 to −1.86)**	**−2.06 (−2.78 to −1.34)**	
	R-MS	**−1.53 (−2.76 to −0.30)**	−0.71 (−1.69 to 0.27)	**1.35 (0.24 to 2.46)**
Irritation	HS	0.07 (−0.65 to 0.80)		
	U-MS	**−1.52 (−2.64 to −0.39)**	**−1.59 (−2.56 to −0.62)**	
	R-MS	**−1.98 (−3.06 to −0.90)**	**−2.05 (−2.96 to −1.15)**	−0.46 (−1.69 to 0.76)
Anxiety	HS	**0.75 (0.07 to 1.43)**		
	U-MS	−0.38 (−1.44 to 0.69)	**−1.12 (−2.07 to −0.18)**	
	R-MS	−0.89 (−1.91 to 0.13)	**−1.63 (−2.51 to −0.76)**	−0.51 (−1.68 to 0.66)
Aggressiveness	HS	**2.67 (1.94 to 3.40)**		
	U-MS	0.07 (−1.11 to 1.25)	**−2.61 (−3.77 to −1.44)**	
	R-MS	**−1.73 (−2.52 to −0.94)**	**−4.40 (−5.16 to −3.64)**	**−1.80 (−2.97 to −0.62)**
Concern	HS	−0.25 (−0.91 to 0.40)		
	U-MS	**−1.05 (−2.03 to −0.07)**	**−0.80 (−1.58 to −0.02)**	
	R-MS	**−1.03 (−2.03 to −0.03)**	−0.78 (−1.57 to 0.02)	0.02 (−1.01 to 1.06)
Fear for you	HS	**0.36 (−0.36 to 1.07)**		
	U-MS	−0.51 (−1.55 to 0.53)	**−0.87 (−1.71 to −0.03)**	
	R-MS	−0.05 (−1.20 to 1.09)	−0.41 (−1.35 to 0.54)	0.46 (−0.72 to 1.64)
Fear for your loved ones	HS	**−0.86 (−1.55 to −0.17)**		
	U-MS	**−1.42 (−2.46 to −0.38)**	−0.56 (−1.39 to 0.27)	
	R-MS	−1.01 (−2.11 to 0.08)	−0.15 (−1.04 to 0.73)	0.41 (−0.74 to 1.55)
Loss of interest in your activities	HS	**1.43 (0.73 to 2.12)**		
	U-MS	−0.11 (−1.28 to 1.06)	**−1.53 (−2.63 to −0.44)**	
	R-MS	−0.39 (−1.51 to 0.73)	**−1.82 (−2.85 to −0.79)**	−0.28 (−1.65 to 1.09)
Depression	HS	**2.21 (1.52 to 2.90)**		
	U-MS	0.78 (−0.42 to 1.97)	**−1.43 (−2.59 to −0.28)**	
	R-MS	**−0.88 (−1.71 to −0.06)**	**−3.10 (−3.86 to −2.33)**	**−1.66 (−2.86 to −0.47)**

CI: confidence interval; Δ: mean difference; HS: High School; MS: Middle School; R-MS: Rural Middle School; U-MS: Urban Middle School.

**Table 4 children-09-00059-t004:** Correlation matrix of the whole sample.

Reaction	Indifference	Disbelief	Irritation	Anxiety	Aggressi-Veness	Concern	Fear for You	Fear for YLO	LOI in Your Activities	Depression
Indifference	1									
Disbelief	0.099 ***	1								
Irritation	0.125 ***	0.243 ***	1							
Anxiety	0.072 **	0.185 ***	0.423 ***	1						
Aggressiveness	0.205 ***	0.208 ***	0.518 ***	0.443 ***	1					
Concern	−0.032	0.249 ***	0.284 ***	0.57 ***	0.25 ***	1				
Fear for you	−0.037	0.241 ***	0.27 ***	0.59 ***	0.323 ***	0.692 ***	1			
Fear for YBO	−0.091 ***	0.214 ***	0.209 ***	0.362 ***	0.157 ***	0.537 ***	0.544 ***	1		
LOI in your activities	0.264 ***	0.129 ***	0.305 ***	0.342 ***	0.434 ***	0.207 ***	0.197 ***	0.115 ***	1	
Depression	0.121 ***	0.114 ***	0.375 ***	0.536 ***	0.537 ***	0.321 ***	0.348 ***	0.145 ***	0.488 ***	1

*** *p* < 0.001, ** *p* < 0.01.

## Data Availability

Authors do not wish to share their data without their permission.

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
