# Peer review of "The Emotional Response to Pandemic of Middle- and High-School Students of an Italian Northern Province: The ERP Study"

_children, 2022, doi:10.3390/children9010059_

Round 1
Reviewer 1 Report
Thank you very much for giving me the opportunity to review this article addressing such a topical and impactful issue. I have the following recommendations to strengthen the article.
Introduction
• The information you provide is of interest. However, I would reconsider looking further into the psychological effects of covid on young people, and why have the variables in your questionnaire been selected and not others. I think the justification for all the variables used in the questionnaire should be justified in more depth.
Materials and Methods
• How was the data collection carried out? Online?
• I believe that including the reliability and validity of the questionnaire you have designed would increase the final quality of the document.
• I find the comparison of rural, urban, middle school and high school students very interesting. However, I think it is necessary to justify why comparing data from adult and adolescent populations.
• How was the adult population sample selected?
Results
• Figure 2 shows the gender differences between all participants and the questionnaire variables. However, the rest of the analyses they carry out do differentiate by age group. Adolescents may have experienced it differently than adults.
Discussion
• Remember the objective of the article and the response to it.
• Line 175-177: Anxiety, panic attacks, irritability, and depression were the most frequently mood disorders reported in these cases [31, 33, 40], by a prevalence of approximately 20% [26]. 20% refers to the general world population? I think you should focus the study.
• Some parts of the discussion seem a bit disconnected. Consider making the wording of this section more fluid.
• Include new references in the discussion
Author Response
Dear collegue, below the answers to your comments. Best regards, the authors
Introduction
- The information you provide is of interest. However, I would reconsider looking further into the psychological effects of covid on young people, and why have the variables in your questionnaire been selected and not others. I think the justification for all the variables used in the questionnaire should be justified in more depth.
ANSWER:
The questionnaire used in the present study was mainly deducted by the review and the meta-analysis published by Panda P.Q. et al. in 2021 (see ref. # 31) where several specific questionnaires were considered and commented. We only added the “aggressiveness“ to the possible responses as this aspect was never investigated in all other studies, but we presumed that it would be relevant when dealing with adolescents. We added this information in the manuscript (see rows 68-71)
Materials and Methods
- How was the data collection carried out? Online?
ANSWER:
Data collection was carried out online via google forms in 96.4% of cases and by paper questionnaires in the remaining 3.6%. We added this information in the paper (see rows 80-81).
- I believe that including the reliability and validity of the questionnaire you have designed would increase the final quality of the document.
ANSWER:
As reported in the Methods section, the questionnaire had been previously validated in a different sample of adolescents during the pre-study phase, and its comprehension reached 100%. Moreover we added the Cronbach’s alpha to quantify the reliability (see row 77).
- I find the comparison of rural, urban, middle school and high school students very interesting. However, I think it is necessary to justify why comparing data from adult and adolescent populations.
ANSWER:
Difference between adults and adolescents’ emotional response was one of the research question of our study. Hence, a pre-calculated sample of adults was included in order to investigate and compare the pattern of response of the two samples of subjects different by their age.
- How was the adult population sample selected?
ANSWER:
In this case, data were randomly collected over the same period from people working in or linked to the schools involved in the study , and from parents and relatives of students who accepted to participate in the study. Not to forget the lock down and limitations active in that period in our Country. As adult controls were sampled from the same environment of adolescents, possible confounding should be reduced. We added this information in the paper (see row 90-92)
Results
- Figure 2 shows the gender differences between all participants and the questionnaire variables. However, the rest of the analyses they carry out do differentiate by age group. Adolescents may have experienced it differently than adults.
ANSWER:
You are right, however such comparison was adjusted for age in order to take into account difference among subgroups (adult, high or middle school). We added such specification in the caption of Figure 2. Moreover, results for each subgroup are reported in the supplementary material and equivalent trends were observed in all the four subgroups.
Discussion
- Remember the objective of the article and the response to it.
ANSWER: According to your suggestion, the aim and the message of the study were recalled in the “Discussion” section (see row 2016-2018)
- Line 175-177: Anxiety, panic attacks, irritability, and depression were the most frequently mood disorders reported in these cases [31, 33, 40], by a prevalence of approximately 20% [26]. 20% refers to the general world population? I think you should focus the study.
ANSWER: As reported in the manuscript, the % mentioned were deduced from specific studies on adolescents and teenagers.
- Some parts of the discussion seem a bit disconnected. Consider making the wording of this section more fluid.
ANSWER: According to your suggestion, some parts of the Discussion have been reworded and implemented
- Include new references in the discussion
ANSWER: According to your suggestion, some references have been added in the discussion (see references 42-44)

Reviewer 2 Report
Dear colleagues, I hope this message find you well.
Thank you for giving me the opportunity of reading the work “The Emotional Response to Pandemic of middle- and high- school Students of an Italian Northern Province: The ERP study”, it has been a very big pleasure to collaborate reviewing this manuscript. The topic of this paper is very interesting and it seems necessary to delve it. However, there are several questions to improve before to publish it:
Introduction
- I recommended to divide the introduction into several subsections in order to do it more intuitive.
- On the other hand, when you explain the signs of psychological distress and mental health as a result of COVID-19 (page 1), it is necessary to add more data. I recommend you to add this paper recently published which has been develop in Italy: https://doi.org/10.3390/ijerph18147422
- Objectives and hypotheses are not clear.
Method
- I recommend adding more information. For example, sample information should be detailed here instead of results section.
Discussion
- It is necessary to describe in more detail the practical and theoretical implications.
- Please, limitations should be explained more deeply.
Other issues
- It is advisable to do a proofreading.Extensive editing of English language and style are required.
Author Response
Dear collegues, thank you for your comments. Below our answers.
Best regards,
the authors
Introduction
- I recommended to divide the introduction into several subsections in order to do it more intuitive.
ANSWER:
Dear collegue, the Introduction was already structured into specific subsection: paragraph 1 (background), paragraph 2 (previous studies in general population), paragraph 3 (previous studies in adolescents), paragraph 4 (focus on Italy) and paragraph 5 (objective of the study)
- On the other hand, when you explain the signs of psychological distress and mental health as a result of COVID-19 (page 1), it is necessary to add more data. I recommend you to add this paper recently published which has been develop in Italy: https://doi.org/10.3390/ijerph18147422
ANSWER: Thank you for the suggestion, we added this paper in the introduction
- Objectives and hypotheses are not clear.
ANSWER: We revised the last paragraph of Introduction
Method
- I recommend adding more information. For example, sample information should be detailed here instead of results section.
ANSWER:
In the Method we explained the sample size calculation, based on preliminary results of plot study. The real number of patients enrolled is a result of the study and should be correctly reported in the Results.
Discussion
- It is necessary to describe in more detail the practical and theoretical implications.
ANSWER: Some suggestions have been added to “Discussion” and “Conclusions”
- Please, limitations should be explained more deeply.
ANSWER: We added more details
Other issues
- It is advisable to do a proofreading. Extensive editing of English language and style are required.
ANSWER: Whole manuscript was revised

Round 2
Reviewer 1 Report
The authors have made modifications to the work, which have improved its final quality.